# Volcanic Ash Resuspension in Patagonia: Numerical Simulations and Observations

**Leonardo Mingari [1,\*], Arnau Folch [1], Lucia Dominguez [2] and Costanza Bonadonna [2]**

1    Barcelona Supercomputing Center, 08034 Barcelona, Spain; afolch@bsc.es
2    Department of Earth Sciences, University of Geneva, 1205 Geneva, Switzerland;
     Lucia.Dominguez@unige.ch (L.D.); Costanza.Bonadonna@unige.ch (C.B.)
\*    Correspondence: leonardo.mingari@bsc.es

**Abstract:** Resuspension of pyroclastic deposits occurs under specific atmospheric and environmental conditions and typically prolongs and exacerbates the impact associated with the primary emplacement of tephra fallout and pyroclastic density current deposits. An accurate forecasting of the phenomenon, to support Volcanic Ash Advisory Centers (VAACs) and civil aviation management, depends on adapting volcanic ash transport and dispersion models to include specific ash emission schemes. Few studies have attempted to model the mechanisms of emission and transport of windblown volcanic ash, and a systematic study of observed cases has not been carried out yet. This manuscript combines numerical simulations along with a variety of observational data to examine the general features of ash resuspension events in northern Patagonia following the 2011 Cordón Caulle eruption (Chile). The associated outcomes provide new insights into the spatial distribution of sources, frequency of events, transport patterns, seasonal and diurnal variability, and spatio-temporal distribution of airborne ash. A novel modelling approach based on the coupling between Advanced Research core of the Weather Research and Forecasting (WRF-ARW) and FALL3D models is presented, with various model improvements that allow overcoming some limitations in previous ash resuspension studies. Outcomes show the importance of integrating source information based on field measurements (e.g., deposit grain size distribution and particle density). We provide evidence of a strong diurnal and seasonal variability associated with the ash resuspension activity in Patagonia. According to the modelled emission fluxes, ash resuspension activity was found to be significantly more intense during daytime hours. Satellite observations and numerical simulations strongly suggest that major emission sources of resuspended ash were distributed across distal areas (>100 km from the vent) of the Patagonian steppe, covered by a thin layer of fine ash. The importance of realistic soil moisture data to properly model the spatial distribution of emission sources is also highlighted.

**Keywords:** volcanic ash resuspension; Cordón Caulle; FALL3D; WRF-ARW

## 1. Introduction

Major explosive volcanic eruptions can inject large amounts of particles and volcanic gases into the atmosphere, resulting in wide areas of the landscape covered by tephra-fallout deposits. Under specific environmental conditions, loose particles from fresh deposits can be easily remobilised by wind. The recurrence of these events represents a collateral hazard derived from the primary volcanic activity, with long-term impacts on health, environment and agriculture. Additionally, the presence of resuspended ash can also affect aviation (e.g., flight cancellation, disruption of airport operations) and road transport networks.

A remarkable example is the 2011 volcanic eruption of Cordón Caulle (CC) in Chile [1,2], characterised by long-lasting plumes strongly influenced by a complex interplay between eruptive style, unsteady wind directions and deposit erosion [3,4]. Due to the recurrent aeolian remobilisation of the tephra-fallout deposits, the aftermath of this eruption encompassed multiple impacts in northern Patagonia, ranging from airport disruptions to severe deterioration of air quality in populated centres [5,6]. For example, an outstanding resuspension outbreak in October 2011, extensively studied through observations and numerical simulations [7,8], reached multiple cities of Argentina and Uruguay, thereby affecting airport operations and leading to multiple flight cancellations. The arrival of volcanic ash at the Buenos Aires metropolitan area (1350 km from the vent) on 16 October 2011 was detected by air quality monitoring stations, with maximum $PM_{10}$ concentrations of almost 700 $\mu g\,m^{-3}$ [7]. During this outbreak, visibility in some Patagonian locations dropped to less than 100 m, equivalent to a severe dust storm [9]. Wind erosion of volcanic deposits was measured using triple Big Spring Number Eight (BSNE) samplers located in strategic sites with varying topography, vegetation, and wind exposure conditions by Panebianco et al. [10]. The authors highlighted the strong influence of the deposit thickness and vegetation cover on the observed mass transport rates (up to 6.3 $kg\,m^{-1}\,day^{-1}$), which were up to two orders of magnitude greater than typical soil erosion rates in this region before the CC eruption. In 2015, resuspension events were exacerbated after the eruption of Calbuco volcano [11–13], which blanketed with fresh volcanic ash a wide area in northern Patagonia.

The first detailed description of impacts due to ash remobilisation in Patagonia was given by Wilson et al. [14] following the 1991 eruption of the Hudson volcano in Chile. Wilson et al. [14] already pointed out that the impact of continuous ash storms in rural communities was severe including multiple consequences such as interruption of human activities, problems for grazing animals, disruption of air traffic and land transport, exacerbated soil erosion, high clean-up costs, and affectation on air quality and human health.

Remarkable resuspension events have also occurred in other regions worldwide, for example, Alaska [15], Iceland [16–18], Japan [19], or Kamchatka [20,21]. For example, a resuspension event of relic volcanic ash from Katmai volcano in the Valley of Ten Thousand Smokes was described by Hadley et al. [15]. The particular conditions required for triggering such episode were examined: dry atmospheric and land surface conditions, strong winds, a complex terrain, a super-adiabatic lapse rate in the lower troposphere, and a strong subsidence inversion.

In most of the cases, natural mechanisms for ash remobilisation depend on atmospheric conditions (e.g., wind, rainfall), land surface state (e.g., soil moisture, vegetation), topography, and specific features of volcanic deposits (e.g., grain size distribution, particle density, particle shape). Only a few experimental studies have focused on the aeolian erosion of volcanic deposits. Early airborne measurements during remobilisation of fresh deposits at Mount St. Helens (U.S.A) in 1980 already showed that volcanic ash can be resuspended even by modest winds, causing an important reduction of visibility [22]. Similarly, wind tunnel experiments carried out by Fowler and Lopushinsky [23] showed that freshly deposited ash can be resuspended at relatively low wind speeds, while consolidation of tephra-fallout deposits resulting from successive cycles of wetting and drying processes, caused a significant increase in the threshold wind velocity of erosion (i.e., the minimum wind speed required to mobilise the soil particles) [9]. More recently, Del Bello et al. [24] conducted wind tunnel experiments under controlled ambient humidity conditions, highlighting that resuspension of smaller particles is hindered at high humidity levels. In parallel, Etyemezian et al. [25] measured also emission rates and erosion thresholds in a humidity-controlled chamber using a small wind tunnel-like device.

This kind of experiments are of paramount importance to dispersion models, which aim at simulating the main processes involved in the life cycle of remobilised ash: emission, atmospheric transport, and ground re-deposition. Specifically, modelling of ash resuspension focuses on wind remobilised fine particulate matter, which can be transported over large distances by aeolian suspension [26]. One of the first attempts to simulate wind-induced resuspension events was that of Leadbetter et al. [17], which used a simple dust emission scheme for the vertical flux of uplifted

material and showed the feasibility of producing forecasts of ash resuspension episodes. Currently, the London Volcanic Ash Advisory Centre (VAAC) provides daily forecasts of resuspended ash to the Icelandic Met Office (IMO) using the Lagrangian particle model NAME [27]. The modelling system based on the coupling between Advanced Research core of the Weather Research and Forecasting (WRF-ARW) and FALL3D models has also been applied to simulate resuspension of both fresh [7,12] and ancient [28] tephra-fallout deposits with promising results. Folch et al. [7] conducted the first attempt to model an outbreak of ash resuspension in Patagonia using three different emission schemes originally derived for mineral dust. This study highlighted the large sensitivity of the most complex emission schemes to uncertainties in model inputs (e.g., primary fallout deposit features, soil moisture) and parameters which are not well constrained can lead to a poor model accuracy.

Previous work on ash resuspension has largely focused on specific outbreaks, such as the 2003 event at Katmai [15], the 2011 event associated with the Cordón Caulle deposit [7,8], the 2013 event associated with the Grímsvötn/Eyjafjallajökull deposits [18] and the 2015 Calbuco event [12]. However, the general features associated with periodic and long-lasting resuspension events have not been addressed in a systematic way. This paper sheds new light on the spatial distribution of emission sources, diurnal and season variability, frequency of events, transport patterns, and spatio-temporal distribution of airborne ash in northern Patagonia following the 2011 CC eruption. This study combines numerical simulations performed with the WRF-ARW/FALL3D modelling system with an observational dataset gathering weather station data, satellite imagery and lidar measurements. A new implementation of the emission scheme in FALL3D aims at overcoming some limitations from previous studies by including: (i) a better characterisation of the tephra-fallout deposit through field-based grain size distribution (GSD) and particle density based on experimental studies [26]; (ii) a new strategy to obtain a more realistic description of the top-layer soil moisture, including a better initialisation of the land surface model; (iii) the effect of vegetation cover on wind erosion, and; (iv) a new approach to define the grain size of resuspended particles. The maximum size for emitted particles depends on atmospheric conditions and particle properties and no arbitrary restriction was imposed on the resuspended particle sizes. The suitability of the proposed modelling strategy to reproduce the observations is discussed in detail.

This study is framed in the context of developing operational forecast capabilities to predict the occurrence of resuspension events in the Andean volcanic region. The final product is intended to provide support to VAACs, air quality agencies and civil aviation management. In particular, the resulting modelling system, based on the coupling of WRF-ARW and FALL3D models, is intended to be adopted as operational model at the Buenos Aires VAAC to forecast ash resuspension in the near future. The manuscript is organised as follows. Section 2 gives an overview of cases of volcanic ash resuspension reported for the southern portion of South America, including major emission sources. The modelling strategy is outlined in Section 3 along with the parameterisations used in the enhanced emission scheme. Section 4 presents numerical results for different events as well as an overview of the different types of available observations. In Section 5, findings are discussed and conceptually integrated. Conclusions are drawn in the final Section 6.

## 2. Background

During the last three decades, remarkable volcanic eruptions have occurred in Patagonia, and a number of well-documented cases of subsequent remobilisation of fallout deposits have been reported. For example, following the 1991 Hudson eruption, volcanic deposits were highly erodible for years in windy zones with scarce vegetation cover [14]. The low rainfall regime in these regions had a double effect: (i) the erosive potential of soil was increased and, (ii) the consolidation process of fresh pyroclastic deposits was delayed. Subsequently, the 2011 Cordón Caulle and 2015 Calbuco explosive eruptions, along with the remobilisation of the resulting fallout deposits, caused severe impacts on environment and disruptions on human activities, which prolonged for several months over dry regions of northern Patagonia [5,12,29]. Additionally, notable episodes of ash remobilisation

related to ancient volcanic deposits are frequently observed in dry regions of north-western Argentina, including areas around the Cerro Blanco volcanic complex and Fiambalá Basin [28]. The mentioned cases represent the most relevant events of ash resuspension reported in the literature for the southern edge of South America. The approximate locations of the major sources of resuspended ash in this region are indicated in Figure 1 (red arrows) with the key volcanoes (red triangles) involved.

The spatial distribution of potential sources of volcanic ash resuspension follows a similar pattern to that found for dust sources. In South America, dust source areas are concentrated along a continuous band of arid and semi-arid terrains extending from the Peruvian coast in the North to Patagonia in the South [30]. Three major dust sources have been identified: the Puna-Altiplano Plateau, central-western Argentina and Patagonia [31,32] (Figure 1). Similarly, ash resuspension occurs predominantly in arid and semi-arid regions, with the most affected areas being characterised by annual precipitations below 250 mm.

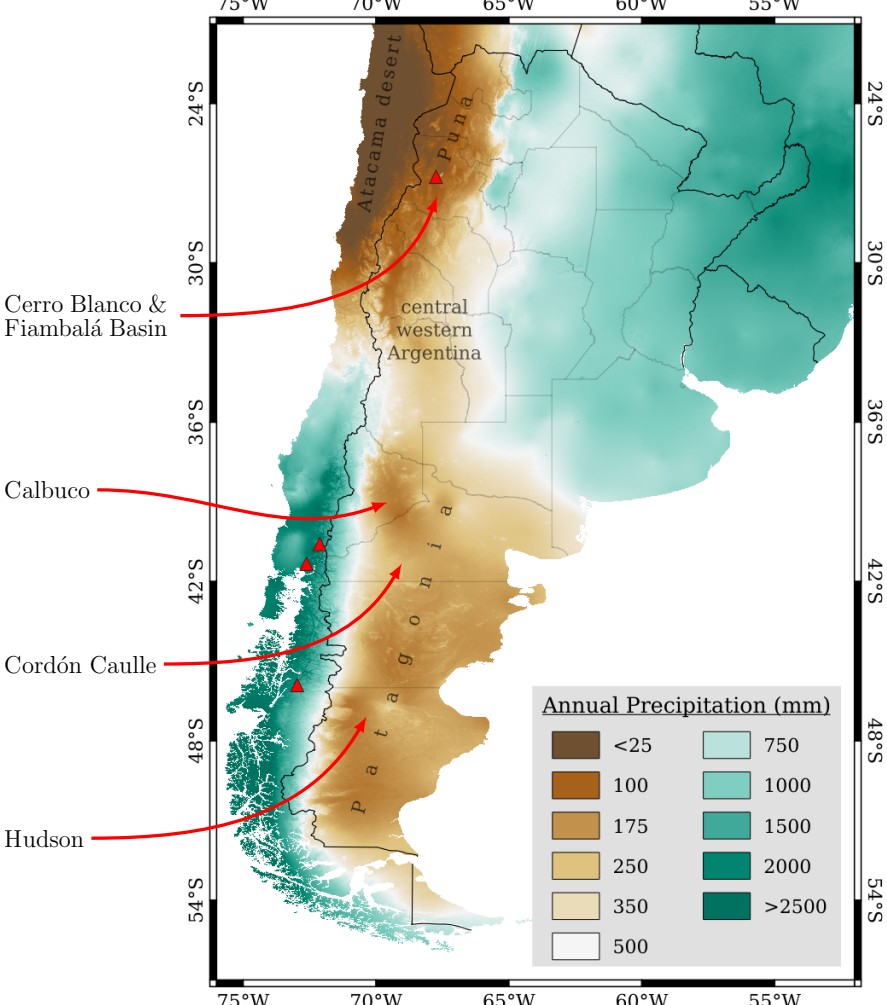

**Figure 1.** Approximated locations for the major ash-resuspension sources reported in the literature for South America (indicated by red arrows). These areas are concentrated along a continuous band of arid and semi-arid terrains. To illustrate this, the geographic distribution of mean annual precipitation according to the climatic database by Fick and Hijmans [33] corresponding to the 1970–2000 time period is shown. The highlighted sources are associated with the eruption of the key volcanoes indicated by red triangles (from bottom to top): Hudson (1991), Calbuco (2015), Cordón Caulle (2011) and Cerro Blanco (~4.2 ka).

The tephra-fallout deposits over the northern region of Patagonia associated with the 2011 CC [1,2] and the 2015 Calbuco [11,12] eruptions are periodically remobilised by wind nowadays.

The Patagonian climate is influenced by two key factors, the strong westerlies and the orographic effect of the Andes range [34]. Indeed, the Andes mountains constitute a barrier which partially blocks the prevailing westerly winds that carry moist air from the Pacific ocean to the continent. As a result, orography-driven precipitation is dominant along the western side, whereas on the eastern slopes of the Andes the mean annual precipitation decreases eastwards down to 200 mm, with a very steep gradient over the Central Patagonian Plateau [35]. In addition, due to the prevailing westerly winds, tephra of the 2011 CC eruption was predominantly transported towards the east and south-east, resulting in a wide area of the arid and semi-arid regions of northern Patagonia severely affected by tephra dispersal and fallout. In contrast, most of the tephra associated with the 2015 Calbuco eruption was transported towards the north-east [36].

According to Craig et al. [37], one of the most notable aspects of the remobilisation phenomena associated with the CC tephra-fallout deposit, was the presence of two distinct impact areas and recovery times. A rapid recovery could be identified in temperate environments associated with a proximal, thick and coarse primary deposit (e.g., Bariloche or San Martín de los Andes shown in Figure 2). Conversely, the severity and duration of remobilisation events was considerably higher on the Patagonian steppe, the semiarid region eastward from the Andes, characterised by grasses and shrubby vegetation. In this region, prevalent windy conditions, a dry climate, and the scarcity of vegetation provided propitious conditions for volcanic ash remobilisation. With these specific environmental conditions, wind remobilisation was prolonged for years [29] affecting several communities in the Patagonian steppe (e.g., Pilcaniyeu, Comallo, Ingeniero Jacobacci, Maquinchao).

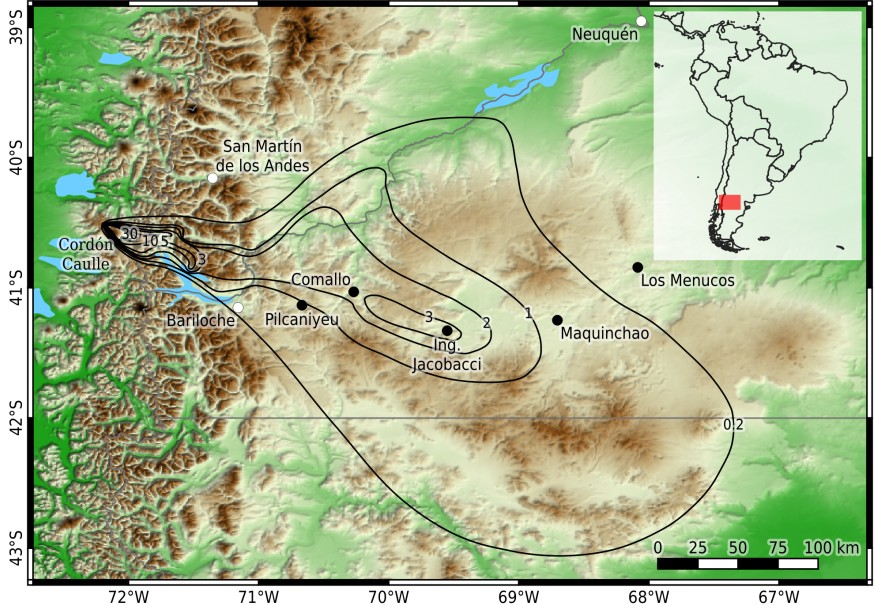

**Figure 2.** Map of the study region with the main localities of the Patagonian steppe affected by the tephra fallout following the 2011 CC eruption. The isopach map of Unit III of the associated tephra-fallout deposit is also indicated with the thickness contours in centimetres (adapted from Dominguez et al. [26]). Elevation data extracted from the Global Multi-resolution Terrain Elevation Data 2010 (GMTED2010) dataset.

## 2.1. Primary Tephra-Fallout Deposit

An accurate description of the primary tephra-fallout deposit distribution and its granulometry are pivotal when modelling the resuspension flux. Located in the Southern Volcanic Zone of the Central Andes, the CC volcano erupted on 4 June 2011 and produced a long-lasting rhyolitic eruption whose plumes reached up to 14 km above the vent. The interaction between the eruptive activity and variable strong winds generated a complex deposit of about 1 km$^3$ constituted by four stratigraphic units [2,3]. Based on field observations of the primary and remobilised deposits, Dominguez et al. [26]

identified the upper mostly fine-grained Unit III, deposited during 6–12 June, as the most susceptible to remobilisation. Figure 2 shows the isopach map of Unit III, compiled by Dominguez et al. [26] by combining thickness measurements of Pistolesi et al. [2] and Gaitán et al. [38]. The horizontal extension of the deposit was validated with the analysis of MODIS (Terra and Aqua) and Landsat satellite images over time (4–21 June 2011, 14 August 2011, 18, 25, 27 October 2011).

One of the most crucial parameters for emission models is the Grain-Size Distribution (GSD) of tephra-fallout deposits. Tephra-fallout deposits are granulometrically heterogeneous, with median grain size typically decreasing with distance from the vent. Complexities related to instabilities in the plume or the occurrence of aggregation phenomena can also lead to strong grain size variations across the deposit [4]. Unit III consists of 5 stratigraphic layers (K1 to K5). In this study we considered 11 proximal samples collected from 6 sampling sites (indicated by black circles in Figure S1, supplementary material) and 4 distal samples (of layers K1 to K4) collected from a single sampling site (indicated by a black star in Figure S1, supplementary material). Proximal samples (<80 km from the vent) correspond to sites distributed in a region delimited by the 3-cm isopach contour (blue contour in Figure S1) and distal samples (∼240 km from the vent) correspond to a site close to Ing. Jacobacci, between the 3-cm and 0.2-cm isopach contours (blue and red contours in Figure S1, respectively). Individual samples of Unit III were used to compute the weighted average GSD from proximal and distal samples. Both proximal and distal average GSD are shown in Figure S2 (supplementary material). The average proximal GSD is characterised by a bimodal distribution, with the mode of the coarse-grained subpopulation at roughly $-1\Phi$ (i.e., 2 mm) and the mode of the fine-grained subpopulation at $\sim 4.5\Phi$ (i.e., 44 μm) for both proximal and distal GSDs. The fine-grained subpopulation is the most relevant to resuspension as particles released from surface remain suspended in atmosphere if $d \lesssim 80\,\mu m$, under typical atmospheric conditions (see Section 3).

To simulate the atmospheric transport of resuspended fine ash, in this work we consider the Dense Rock Equivalent (DRE) density for all particles classes. This particle density, taken as the skeletal density without open vesicularity, was measured using a helium pycnometer at the University of Geneva. The average density of remobilised particles was found to be $2490 \pm 10\,kg\,m^{-3}$.

## 3. Materials and Methods

The modelling system aims at representing the main processes involved in the life cycle of remobilised ash, that is, emission, atmospheric transport, and deposition. The modelling strategy consists of two consecutive steps coupling the WRF-ARW and the FALL3D models in offline mode. In the first step, the WRF-ARW model (see Section 3.2) is run to obtain the meteorological fields required to drive the FALL3D dispersal model. Subsequently, the advection, diffusion and deposition of remobilised ash are simulated by the FALL3D model version v7.3 [39–41]. In this study, a new implementation of the volcanic ash resuspension module of the FALL3D model is presented.

### 3.1. Volcanic Ash Resuspension Module

The mobilisation by wind of soil particles depends on the transfer of momentum from the atmosphere to the rough ground surface, which can be quantified through the surface Reynold's stress:

$$\tau = \rho_a u_*^2, \tag{1}$$

where $\rho_a$ is the air density and $u_*$ is the so-called friction velocity, a scaling velocity defined in terms of the covariance of the fluctuations of the horizontal and vertical wind components $(u'_x, u'_y, u'_z)$ [42], that is,

$$u_* = \left( \overline{u'_x u'_z}^2 + \overline{u'_y u'_z}^2 \right)^{1/4}. \tag{2}$$

The variation in shear stress for a wall-bounded shear flow is typically negligible close to the wall in a well-developed turbulent boundary layer. As a consequence, the total shear stress can be assumed

constant and equal to $\tau_w$, the local shear stress at the wall [43]. As the laminar contribution to the total shear stress can be neglected in this case (except within the viscous sublayer), Equation (1) becomes:

$$\tau_w \approx \rho_a u_*^2. \tag{3}$$

This relationship actually explains why most of the emission schemes for mineral dust are parameterised in terms of the friction velocity [44–48]. In general, experimental data shows a sharp increase of dust production with wind speed and friction velocity [49] and, typically, simple emission schemes assume an asymptotic behaviour for the vertical flux of particles given by

$$F \sim u_*^n. \tag{4}$$

For instance, Westphal et al. [50] proposed the following expression for the total emission flux:

$$F = 23 \times 10^{-5}\, u_*^4, \tag{5}$$

with $u_*$ in $\mathrm{m\,s^{-1}}$ and $F$ in $\mathrm{kg\,m^{-2}\,s^{-1}}$.

The new implementation in the FALL3D model of the volcanic ash emission scheme based on the Shao et al. [51,52] approach is sketched in Figure 3.

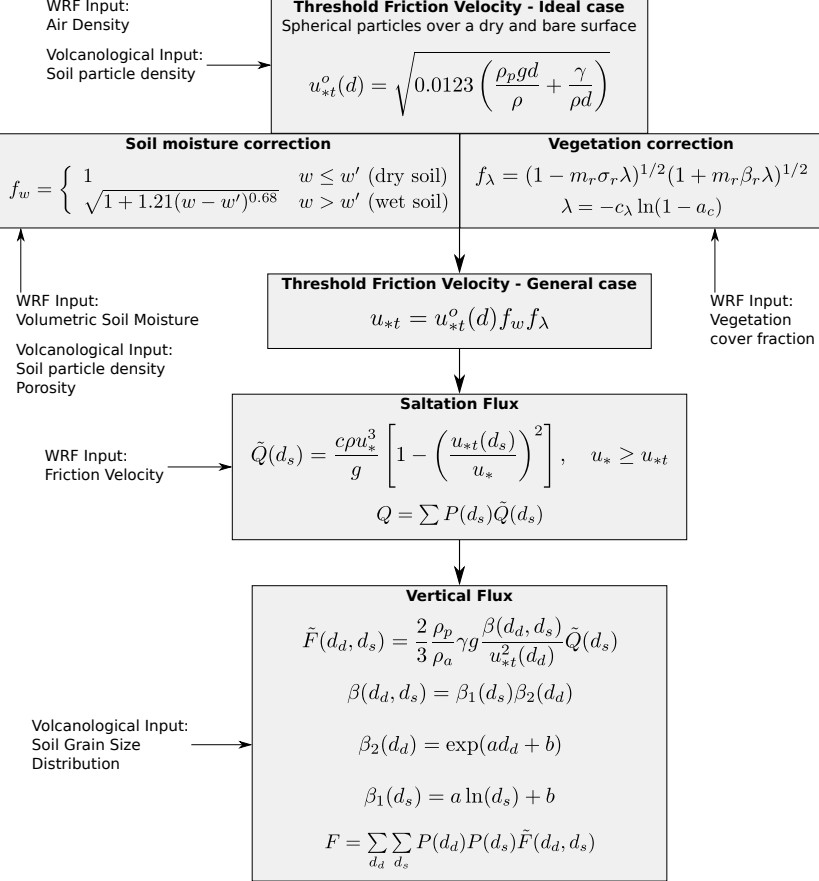

**Figure 3.** Implementation of the volcanic ash resuspension module in FALL3D based on the original dust emission scheme by Shao et al. [51,52]. The parameterisations and the required input data are detailed along with the procedure to compute the emission fluxes.

The threshold friction velocity ($u_{*t}$) is the minimum friction velocity required to initialise the movement of soil particles, and represents the resistance of surface against the wind erosion. Here, the expression derived by Shao and Lu [53] for spherical particles loosely spread over a dry and

bare surface was used to compute $u_{*t}$. The dependency of $u_{*t}$ on particle size for typical values of the experimental parameter $\gamma$, which takes into account the effect of inter-particle cohesion, is shown in Figure 4 (black lines). This parameter was assumed to be $\gamma = 3 \times 10^{-4}$ kg s$^{-2}$ in previous works simulating the remobilisation of ancient volcanic deposits [28]. However, according to recent field observations presented by Dominguez et al. [54], particles with sizes ranging between 65 and 80 µm are the most abundant particles in the secondary deposit associated with remobilised material, suggesting that $\gamma$ should probably be less than $\gamma = 3 \times 10^{-4}$ kg s$^{-2}$. Consequently, this work assumes $\gamma = 1.65 \times 10^{-4}$ kg s$^{-2}$, that is, the minimum value within the experimental range found by Shao and Lu [53]. Moreover, this supposition is consistent with the intuitive idea that fresh and loose volcanic deposits will be more easily eroded by wind, resulting in lower erosion thresholds associated with low $\gamma$ values.

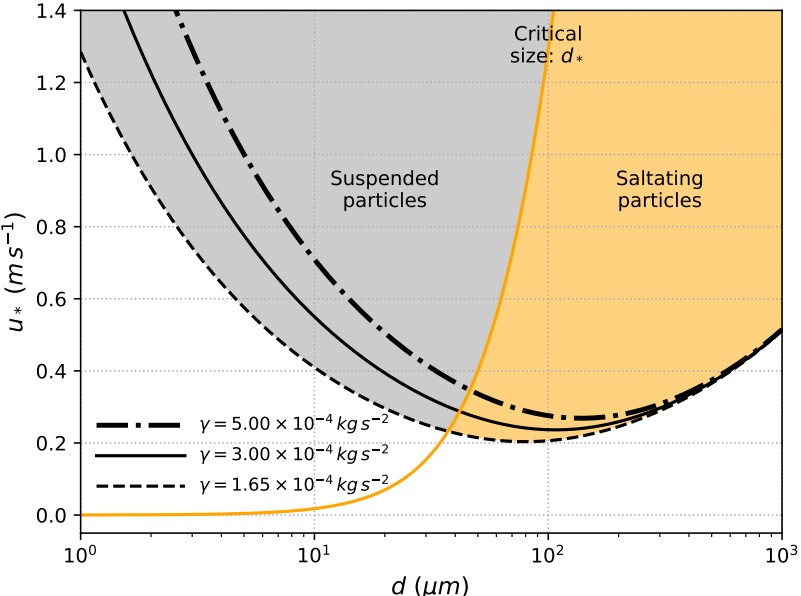

**Figure 4.** Two-dimensional diagram in the $u_*$–$d$ space (i.e., friction velocity and particle size) showing the conditions required to initiate suspension and saltation of particles from a state of repose. Curves of static threshold friction velocity for spherical particles loosely spread over a dry and bare surface are computed according to Shao and Lu [53] model (black lines). For illustrative purposes, typical values of the experimental parameter $\gamma$, accounting for the effect of inter-particle cohesion, are considered. In this work, $\gamma = 1.65 \times 10^{-4}$ is assumed. The critical particle size, $d_*$, is used in this work to define the transition, established via the condition given by Equation (9), between the saltation and suspension types of particle movement.

The correction factors $f_w$ and $f_\lambda$ are reduction factors accounting for the effect of soil moisture and vegetation on the threshold friction velocity, respectively (see Figure 3). For the moisture correction $f_w$, the empirical expression obtained by Fécan et al. [55] is considered. This factor depends on the gravimetric soil moisture, $w$, and maximum amount of adsorbed water, $w'$. Figure 5 shows $f_w(w)$ for different values of $w'$ (the value $w' = 1\%$ used by Folch et al. [7] is assumed throughout this work). Note that, for typical ranges of values of $w$ found in arid and semi-arid regions, the soil moisture correction $f_w$ can be highly sensitive to changes in $w$. For the vegetation correction $f_\lambda$, the model uses the theoretical expression derived by Raupach [56] for rough surfaces. Here, the typical values for the experimental parameters are assumed to be $\beta = 90$, $\sigma = 1$, and $m = 0.5$, which produce a good fit to experimental data according to Raupach et al. [57]. The effect of vegetation is taken into account by using the following expression relating the roughness density $\lambda$, and the vegetation cover fraction $a_c$:

$$a_c = 1 - \exp(-\lambda/c_\lambda), \tag{6}$$

where the parameter $c_\lambda = 0.35$ has been found to give a reasonable agreement between predictions and measurements [52]. The dependency of $f_\lambda$ on vegetation cover fraction is also shown in Figure 5 (red curves).

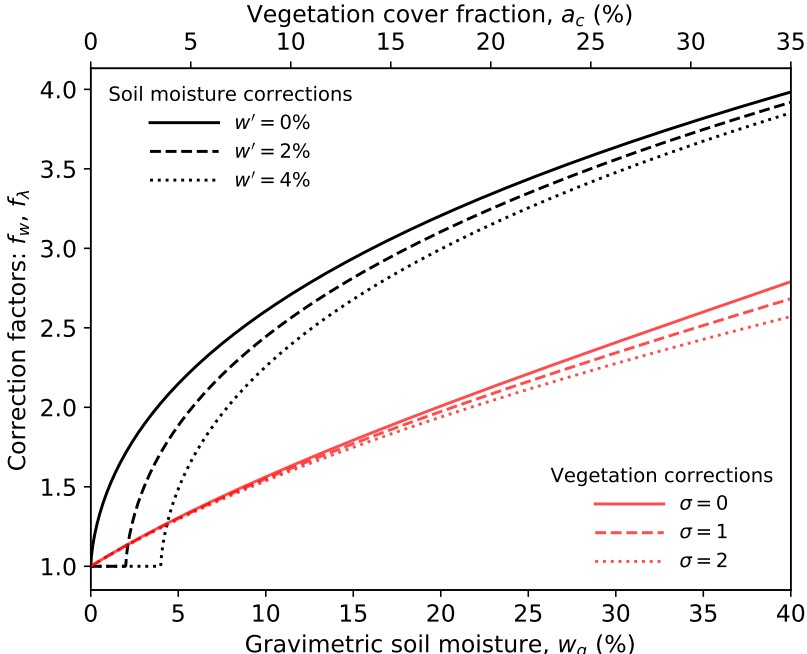

**Figure 5.** Correction factors for the threshold friction velocity accounting for the effects of soil moisture and vegetation. The factor $f_w$ depends on the gravimetric soil moisture, while the factor $f_\lambda$ depends on the vegetation cover fraction. The wind erosion threshold, $u_{*t}$, is increased due to the effect of both factors.

The vertical flux of resuspended ash is computed from the physically-based emission scheme for mineral dust by Shao et al. [51,52]:

$$\tilde{F}(d_d, d_s) = \frac{2}{3} \frac{\rho_p}{\rho_a} \gamma g \frac{\beta(d_d, d_s)}{u_{*t}^2(d_d)} \tilde{Q}(d_s),\tag{7}$$

where $\tilde{F}(d_d, d_s)$ represents the vertical flux of dust particles of size $d_d$ ejected by bombardment of saltating grains of size $d_s$; and $\tilde{Q}(d_s)$ is the vertically integrated stream-wise saltation flux computed using the expression given by Owen [58]. For volcanic deposits with multiple particle sizes, we assume a grain size distribution discretised in bins of size $d$ contributing to the total mass with a fraction $P(d)$. In this case, total vertical flux of resuspended particles can be expressed as [59]:

$$F(d_d) = \sigma P(d_d) \sum_{d_s} P(d_s) \tilde{F}(d_d, d_s),\tag{8}$$

where $F(d_d)$ is the total flux associated with suspended particles of size $d_d < d_*$; and $\sigma$ is a global erodibility parameter. In this work, $\sigma$ represents a calibration factor which is assumed to be independent of position; and it is essentially a reduction factor ($\sigma < 1$) for the emission flux that accounts for the wind-erodible fraction of soil. By comparison of observations and numerical simulations, the optimal value of $\sigma$ was found to be $\sigma = 0.21$, as described in Section 4.3.2.

The sum in Equation (8) accounts for the contribution corresponding to all groups of saltating particles with sizes $d_s > d_*$, being $d_*$ a critical size that defines an ideal limit between saltating and suspended grains. Following Scott et al. [60], the transition condition is established by:

$$v_{set}(d_*) = \kappa u_*, \tag{9}$$

where $v_{set}$ is the settling velocity and $\kappa = 0.4$ is the von Karman constant. We obtained a numerical solution of Equation (9), with the settling velocity being computed as in Ganser [61], in order to determine the critical size $d_*$ between saltating and suspended particles as a function of friction velocity. The results are presented in a 2-dimensional phase diagram showing the boundary conditions between saltation and suspension motion mechanisms (Figure 4).

In this work, by ash remobilisation we mean windblown particles of volcanic origin, regardless of the transport mechanism involved. On the other hand, ash resuspension refers to fine particulate matter with size $d < d_*$, which can be transported over large distances by aeolian suspension. It is important to point out that the maximum size of particles that are potentially resuspended is not fixed, as the transition size $d_*$ is dynamically obtained depending on meteorological conditions (through the friction velocity) and the density/shape of particles (through the settling velocity). For atmospheric conditions prevailing during resuspension events (e.g., $u_* \approx 0.5 \, \mathrm{m \, s^{-1}}$), the transition size roughly corresponds to the maximum size typically attributed to fine ash, that is, $d_* = 62.5 \, \mu\mathrm{m}$. However, according to our modelling strategy, coarse ash can also be resuspended for winds strong enough to satisfy condition $v_{set} < \kappa u_*$. Furthermore, the transition size in Figure 4 is computed assuming a reference particle density for dust ($\rho = 2650 \, \mathrm{kg \, m^{-3}}$), but larger ash particles could also be resuspended for lower density values. As a final remark, it should be noted that the parameterisation given by Shao et al. [52] for the experimental parameter $\beta(d_d, d_s)$ in Equation (7) is only valid for $d_s > 76 \, \mu\mathrm{m}$. In consequence, the summation in Equation (8) is intended over all $d_s$ satisfying $d_s > \max(d_*, 76 \, \mu\mathrm{m})$ when computing the flux $F(d_d)$.

### 3.2. WRF-ARW Simulations

The ARW (Advanced Research WRF) core of the WRF (Weather Research and Forecasting) model [62] was used in this work to obtain the meteorological fields required to drive the dispersal model (e.g., wind speed); and to compute the emission flux for resuspended ash (e.g., friction velocity, soil moisture). The model domain was set up as a 2-way nest, with the outer domain covering the southern portion of South America and the inner domain covering northern Patagonia. For the atmospheric fields, initial conditions for both domains and boundary conditions for the outer domain were defined using 6-h global data from the ERA-Interim dataset with spatial resolution of 0.75° and 60 vertical levels [63].

A realistic representation of soil moisture is of great importance to properly represent the spatial distribution of emission sources for ash resuspension, as discussed in Section 4. In consequence, a different approach was adopted for the initial conditions of the WRF-ARW land-surface model (LSM), as described in detail below. The representation of land processes in WRF is achieved by coupling an atmospheric model with a land surface scheme, which provides heat and moisture fluxes over land surface to the parent atmospheric model. In particular, the soil moisture is predicted solving a diffusion-type equation for the water transport in the soil. The Noah LSM with 4 soil layers depth of 10, 30, 60, and 100 cm, from top to bottom, was used in previous works to simulate volcanic ash resuspension [7,28]. However, the soil moisture for the upper soil layer of 10-cm depth appears to be unrepresentative of the soil surface conditions involved in dust/ash emission processes (for further discussion, see Reference [64]). For this reason, the land surface is modelled here with the 9-level Rapid Update Cycle (RUC) LSM, which provides a more detailed description of the upper soil layer [65,66]. It has nine vertical levels in soil, ranging from the soil surface to 300 cm (i.e., 0, 1, 4, 10, 30, 60, 100, 160, 300 cm), with higher resolution near the interface with the atmosphere.

A sequence of daily numerical simulations from 1 to 30 October was conducted using the WRF-ARW model initialised at 00:00 UTC, with a 48-h run for each day in this period of time (results are presented in Section 4.2). The horizontal resolutions of the 2-way nested domains were 24 and 8 km. A diagram of the modelling approach is depicted in Figure 6.

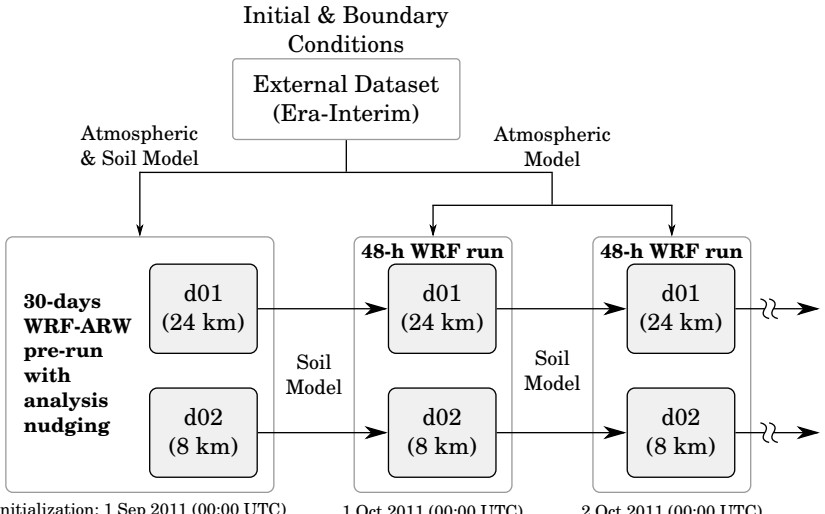

**Figure 6.** Schematic diagram of the sequence of daily WRF-ARW simulations conducted to reconstruct a complete time series of relevant variables during October 2011. Initial and boundary conditions for atmospheric fields were defined from ERA-Interim dataset. The land-surface model for the first run of the sequence was initialised from a soil spin-up run with analysis nudging. The subsequent runs were initialised from the precedent 24 h forecast in the sequence.

Special attention was paid to the strategy required to initialise the land-surface model with stabilised initial conditions for the soil variables. A proper initialisation of the RUC LSM requires soil data at specific levels, which is not provided by global datasets (e.g., GFS, ERA-Interim or ERA-5). In order to get an equilibrated land surface state, we performed a 30-days soil spin-up run with analysis nudging [67,68] starting on the 1st September 2011 at 00:00 UTC. The initial conditions for soil variables in the first run of the sequence (1 October) were then obtained from the 30-days soil spin-up run (see Figure 6). The importance of a proper initialisation of soil moisture was highlighted by Angevine et al. [69], who also considered a spin-up period of 1 month for the land surface model. The effect of this soil initialisation approach is evident if one compares the upper soil moisture obtained directly from the ERA-Interim dataset (Figure 7a) with that resulting from the LSM having been spun up (Figure 7b). Note that the differences are especially large over the dry region, east of the Andes mountain range (Patagonian steppe), where much drier conditions (soil moisture ranging from 5 to 10 m$^3$/m$^3$) and a more marked gradient from west to east can be seen for the properly initialised LSM.

Section 4.3 focuses on three exceptional outbreaks of ash resuspension. For each case, a WRF-ARW simulation was performed with horizontal resolutions of 18 and 6 km for the 2-way nested domains. Similarly, the RUC LSM was used considering a spin-up period of 30 days. A summary of the WRF-ARW configuration for each test case is given in Table 1.

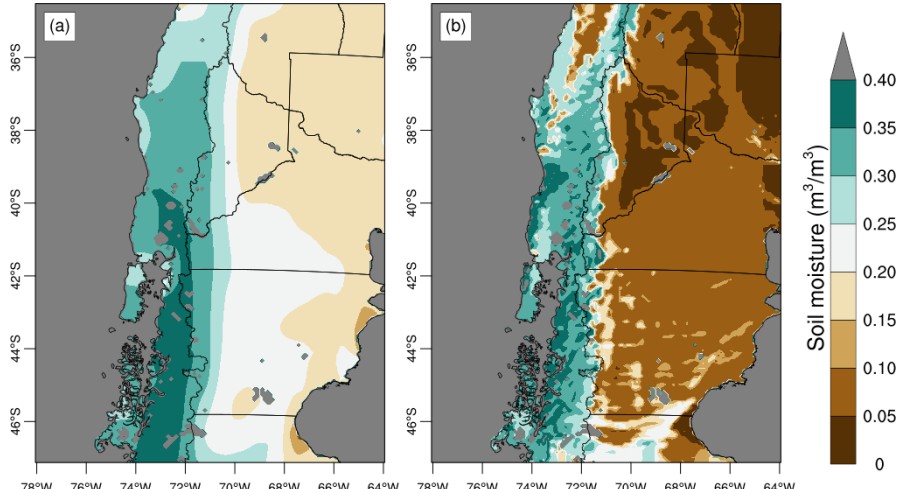

**Figure 7.** Initial fields of upper soil moisture using two WRF-ARW configurations. (**a**) Rapid Update Cycle (RUC) land-surface model initialised directly from ERA-Interim re-analysis data; and (**b**) RUC land-surface model considering a 30-days period for soil spin-up. In both cases, results correspond to the inner WRF domain with horizontal resolution of 8 km.

**Table 1.** Configuration of numerical simulations.

| Start date | Start Time (UTC) | Duration | Horizontal Resolution | Input Data |
|---|---|---|---|---|
| *WRF-ARW* | | | | |
| 26 August 2011 | 12:00 | 84 h | 18/6 km | ERA-Interim [†] |
| 23 September 2011 | 12:00 | 60 h | 18/6 km | ERA-Interim [†] |
| 14 October 2011 | 00:00 | 78 h | 18/6 km | ERA-Interim [†] |
| 16 October 2011 | 12:00 | 36 h | 18/6 km | ERA-Interim [†] |
| *Emission flux* | | | | |
| $\varnothing$ [‡] | $\varnothing$ [‡] | $\varnothing$ [‡] | 0.01° | WRF (6 km) |
| *FALL3D* | | | | |
| 27 August 2011 | 00:00 | 72 h | 0.1° | WRF (18 km) |
| 24 September 2011 | 00:00 | 48 h | 0.1° | WRF (18 km) |
| 14 October 2011 | 12:00 | 84 h | 0.1° | WRF (18 km) |

[†] Land-surface model initialised from a 30-days spin-up run; [‡] Same as the FALL3D simulations.

## 4. Results: The Example of the 2011 Cordón Caulle Eruption

A remarkable consequence of the 2011 CC eruption was the syn- and post-eruptive aeolian remobilisation of fallout deposits, which generated severe impacts on local communities as well as over areas far away from the eruptive centre for several months, prolonging and exacerbating the initial impact of the eruption [5,26,29]. In this section, we present results of numerical simulations and relevant observations. Diurnal and seasonal variability of ash resuspension activity are discussed in Section 4.1 using the frequency of ash-in-suspension events observed at Bariloche. Results of WRF-ARW simulations for October 2011 are presented in Section 4.2 and timeseries of relevant variables are compared to hourly reports issued by the Bariloche weather station. Finally, in Section 4.3 three outstanding outbreaks are simulated using the WRF-ARW/FALL3D modelling strategy.

### 4.1. Seasonal and Diurnal Variability

Ash resuspension activity exhibits strong diurnal and seasonal cycles in northern Patagonia. In recent workshops on wind-remobilisation processes of volcanic ash and associated impact [70,71], local population from the Patagonian steppe have underlined the strong diurnal activity related to ash remobilisation, causing interruption of human activities in the afternoon due to an increased concentration

of airborne ash. Additionally, a seasonal pattern in the distribution of resuspension events along the year was also highlighted by local farmers in the interviews conducted by Forte et al. [29].

The characterisation of resuspension episodes in Patagonia is hindered by the scarcity of meteorological stations and by the lack of ground-based measurements of particulate matter. Nonetheless, hourly reports of ash or dust in suspension from weather stations (e.g., METAR or SYNOP) can provide useful information to analyse the distribution and frequency of events. For example, Shao and Wang [72] reconstructed a climatology of Asian dust events using the following definition for the frequency of events:

$$f_E = \frac{N_E}{N_{obs}},\tag{10}$$

$N_E$ being the number of observed dust-in-suspension events and $N_{obs}$ the total number of reports. This approach was used to estimate the monthly frequency of ash events from reports issued by the weather station at Bariloche, one of the localities most strongly affected by ashfall (see Figure 2). Results for the 2011–2016 time period are presented in Figure 8a. A clear seasonal pattern can be identified in the resuspension activity between 2011 and 2014, with a higher frequency of events occurring in austral spring and beginning of summer (shaded area), in coincidence with the dry and windy season for this region. In contrast, almost no events of ash in suspension can be found in the reports issued during the wet season (austral winter). A sudden increase in the number of events occurred in April 2015 due to the Calbuco eruption, leading to a worsening of the previous situation. However, the regions most severely affected by the Calbuco eruption were located further north [36] and Bariloche recovered quickly. The highest frequency of events occurred in October 2011 (around 4–5 months after the climactic phase of the CC eruption), when ash/dust in suspension was observed and recorded in almost 50% of the reports. This maximum progressively decreased in successive years, as clearly seen in Figure 8a.

The declining trend depends on two key factors: deposit depletion and deposit consolidation. Generally, the recovery time is expected to be closely related to the depletion rate, as wind erosion progressively removes the loose and unconsolidated material from the deposits. However, in the case of CC, temperate environments combined with proximal and thicker deposits showed a rapid recovery, whereas the severity and duration of wind remobilisation was much larger over the Patagonian steppe although ashfall thicknesses were much lower [37]. This suggests that the stabilisation and consolidation of tephra-fallout deposits during the wet season played an important role in preventing ash remobilisation in Bariloche. In fact, wind tunnel experiments have shown that freshly deposited ash can be resuspended at relatively low wind speeds, while successive cycles of wetting and drying processes result on the consolidation of deposits and, therefore, on a significant increase of the erosion threshold [23].

The hourly frequency of events for October and September 2011 is plotted in Figure 8b, where the diurnal variation in the ash resuspension activity becomes evident. The maximum frequency of events occurred in the afternoon at 19:00 UTC (16:00 LT), when the presence of particles suspended in atmosphere was indicated in almost the 70% of the reports corresponding to the September–October 2011 period.

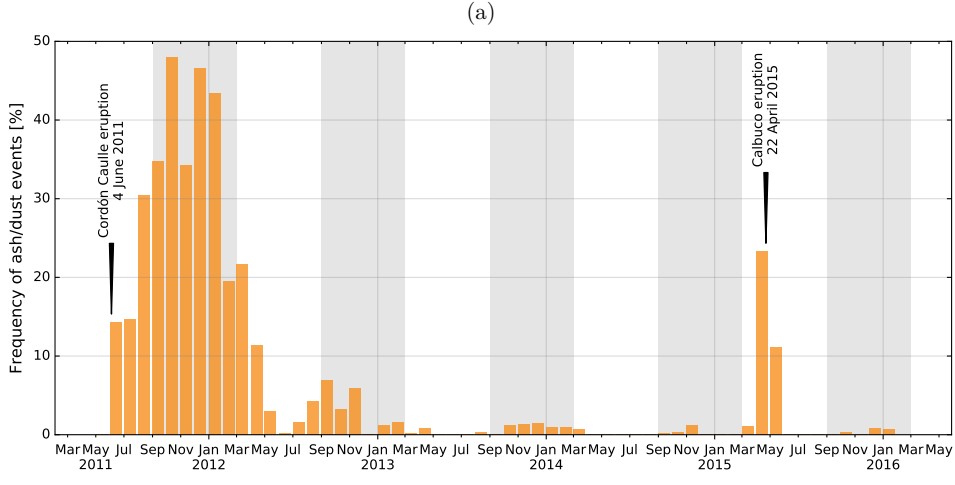

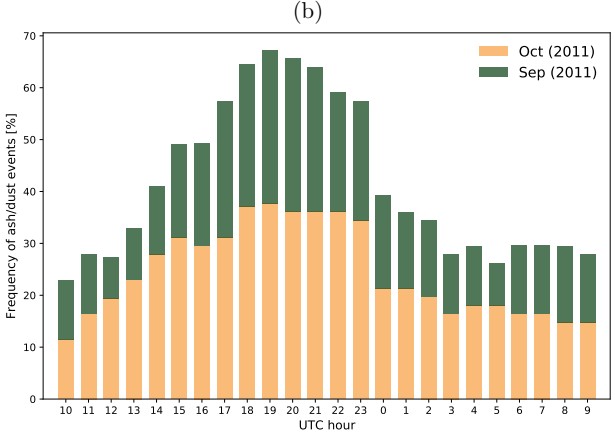

**Figure 8.** Frequency of events of ash/dust in suspension, that is, $f_E$ as defined in Equation (10), estimated from hourly METAR reports issued by the Bariloche weather station. (**a**) Monthly frequency shows a clear seasonal pattern with maximum activity in austral spring (Sep-Oct-Nov) and summer (Dec-Jan-Feb), indicated by shaded areas. Consolidation and depletion of volcanic deposits results in a year-to-year decrease of frequency, $f_E$. New ash resuspension events following the Calbuco eruption are obvious from the new peak in 2015. The onset of Cordón Caulle (4 June 2011) and Calbuco (22 April 2015) eruptions are indicated by arrows. (**b**) Hourly frequency distribution shows a clear diurnal pattern, with most of the events occurring during the afternoon, with the peak of frequency at 19:00 UTC (16:00 LT).

*4.2. Timeseries for October 2011*

Time series of the WRF-ARW variables required by the ash emission scheme (i.e., friction velocity and soil moisture) are plotted in Figure 9. Additionally, Figure 9a shows the times periods with reports of rain (blue shaded area) and suspended ash (green shaded area) by the Bariloche weather station along with model prediction of local ash resuspension (red shaded area), that is, periods of time when $u_* > u_{*t}$. For illustrative purposes, Figure 9d shows the emission flux of resuspended ash computed from Equation (5), corresponding to the simple emission scheme by Westphal et al. [50] (see Section 3.1). A simple expression for the threshold friction velocity is assumed taking into account the effect of soil moisture:

$$u_{*t} = u_{*t}(dry)f_w(w), \tag{11}$$

where the threshold friction velocity for dry conditions $u_{*t}(dry)$ is considered constant here; and $f_w(w)$ is a time-varying correction factor for soil moisture computed using the formula proposed by Fécan et al. [55] (see Figure 3). Results for threshold friction velocity are shown in Figure 9b using $u_{*t}(dry) = 0.24\,\mathrm{m\,s^{-1}}$, obtained below (Figure 10).

According to the simulated emission flux, ash resuspension takes place predominantly during the daytime following a sequence of diurnal peaks, as depicted in Figure 9d (grey shaded areas). The strong diurnal variability in the ash resuspension activity results from two effects fostering emission during the daytime. First, $u_{*t}$ oscillates around $0.35 \, \mathrm{m \, s^{-1}}$ in dry conditions as a consequence of the diurnal variation of soil moisture (Figure 9c). Second, a more marked variability can be recognised in the $u_*$ time series (Figure 9b). Additionally, this simple model reproduces a notable peak on 15 October (Figure 9c), related to the most outstanding event of ash resuspension [7,8].

Note how ash resuspension is inhibited (according to both model predictions and weather station reports) between 6 and 9 October due to rainfall events occurred during this period of time. Simulated time series show a strong increase in soil moisture values (close to the field capacity limit) during this period, which resulted in high threshold friction velocities preventing resuspension even a few hours after the precipitation events.

An assessment of the emission model performance was carried out using the binary classification of events shown in Figure 9a. To this purpose, we computed the percentage of correct predictions, that is, the sum of the fraction of positive ash resuspension events correctly predicted ($f_+$) plus the fraction of negative events correctly predicted ($f_-$):

$$PC = f_+ + f_-. \tag{12}$$

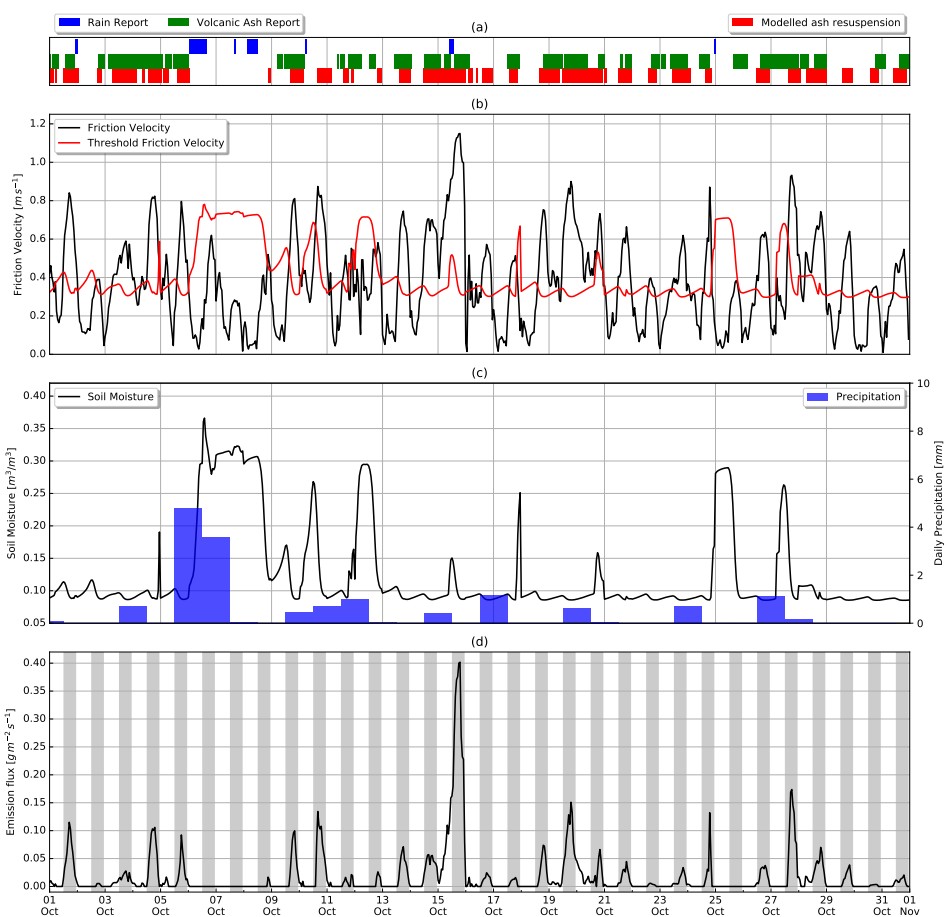

**Figure 9.** WRF time series and ash-in-suspension reports during October 2011 at Bariloche. (**a**) Reports of rain and ash issued by the Bariloche weather station (blue and green shaded areas, respectively) together with hourly predictions of local ash resuspension (red shaded area) according to the model, that is, $u_* > u_{*t}$. (**b**) Friction velocity and threshold friction velocity. (**c**) Soil moisture and daily precipitation. (**d**) Emission flux given by Equation (5). Shaded areas highlight periods of daylight hours between 12:00 and 23:00 UTC (09:00–20:00 LT), when emission predominantly takes place.

This metrics can be further improved using the Heidke skill score (HSS), which also takes into account the number of correct random predictions [73]:

$$HSS = \frac{PC - E}{1 - E},$$
(13)

where $HSS \leq 1$ and $E$ is the probability of a correct prediction purely due to random chance. A value of the Heidke skill score around 0 means no skill, while a perfect forecast obtains a HSS of 1. Figure 10 shows both metrics (PC and HSS) for different values of the parameter $u_{*t}(dry)$. The maximum skill is obtained for an optimal value $u_{*t}(dry) = 0.24\,\mathrm{m\,s^{-1}}$, corresponding to a portion correct and a Heidke skill score of almost $PC = 70\%$ and $HSS = 0.4$, respectively. Note that the optimal dry threshold velocity is consistent with the range of values of $0.2 - 0.25\,\mathrm{m\,s^{-1}}$ given by Shao and Lu [53] for spherical particles loosely spread over a dry and bare surface. However, once the threshold friction velocity is corrected by soil moisture effect using Equation (11), it results on slightly larger values, as can be verified in the time series shown in Figure 9b.

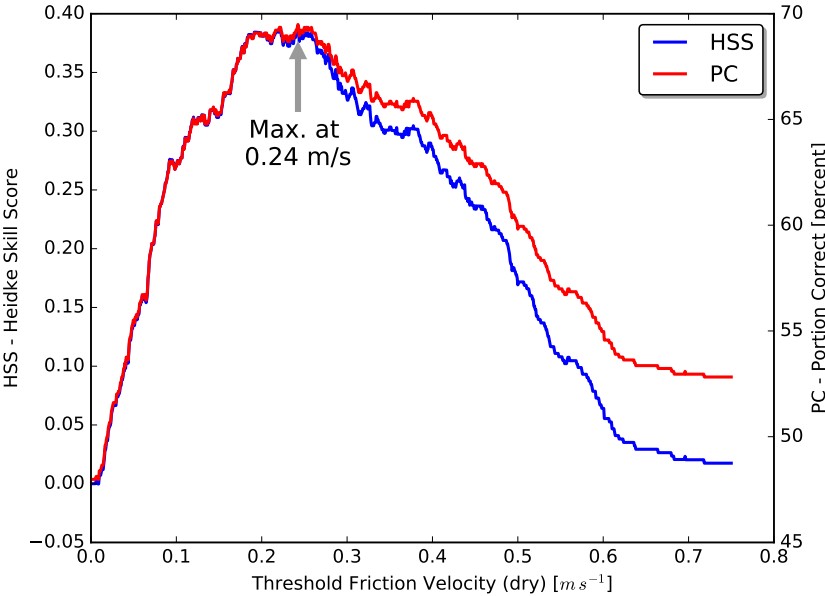

**Figure 10.** Percentage of correct predictions (*PC*) and Heidke skill score (*HSS*) for different values of the parameter $u_{*t}(dry)$. The maximum skill is obtained for the optimal value $u_{*t}(dry) = 0.24\,\mathrm{m\,s^{-1}}$, corresponding to scores of almost $PC = 70\%$ and $HSS = 0.4$.

*4.3. FALL3D Ash Dispersal Simulations*

Three exceptionally strong outbreaks, characterised by long-range ash transport, have been considered to validate the WRF-ARW/FALL3D modelling strategy. Configuration details are provided in Table 1 for each numerical simulation. Ongoing ash resuspension from these events is clearly visible from MODIS satellite imagery on 29 August (Figure 11a), 24 September (Figure 11b), and 15 October (Figure 11c) at 18:15, 18:50, and 19:10 UTC, respectively. In general, satellite detection of resuspended ash is challenging due to the low altitude at which resuspension clouds typically occur [74]. In addition, resuspension outbreaks in Patagonia can be associated with the passage of low pressure systems causing sustained high winds and bringing abundant cloudiness, which make satellite detection more difficult [75]. For these reasons, special attention was paid when selecting these case studies, all characterised by cloud-free conditions and high-quality satellite imagery.

### 4.3.1. Emission Scheme

The vertical mass flux was computed using the emission scheme proposed by Shao et al. [52] described in Section 3.1 (see also Figure 3) using the atmospheric and land-surface fields provided by WRF-ARW (see Section 3.2). For illustrative purposes, Figure 12 shows the key input data required by the emission scheme, that is, 6-km WRF fields for: friction velocity, soil moisture, and vegetation cover fraction (results for 24 September 2011 at 19:00 UTC, the time with most intense wind speeds during this outbreak). The 10-m wind vector field (Figure 12a) shows westerly winds leading to the typical zonal pattern of transport associated with resuspended ash events in this region. It is notable how the orographic barrier, defined by the Andes mountain range, strongly influences the spatial distribution of winds. Ideal conditions for ash resuspension are found downwind of the Andes over the Patagonian steppe, where the highest values of friction velocities can be found.

The spatial distributions of volumetric soil moisture (Figure 12b) and vegetation cover fraction (Figure 12c) exhibit an abrupt transition along the mountain range with dry conditions and scarce vegetation dominating over the steppe region, that is, from the eastern flank of Patagonian Andes to the Atlantic coast. This transition defines two distinct regions for aeolian erosion with contrasting environmental conditions: a wet region where erosion is strongly inhibited and an arid or semi-arid region where ideal conditions for resuspension occur. During the October and September outbreaks, special dry conditions with volumetric soil moisture ranging from 4% to 8% (see Figure 12b) favoured the emission across the steppe region. In contrast, the August outbreak represents an interesting case of resuspension coeval with precipitation events, as described below.

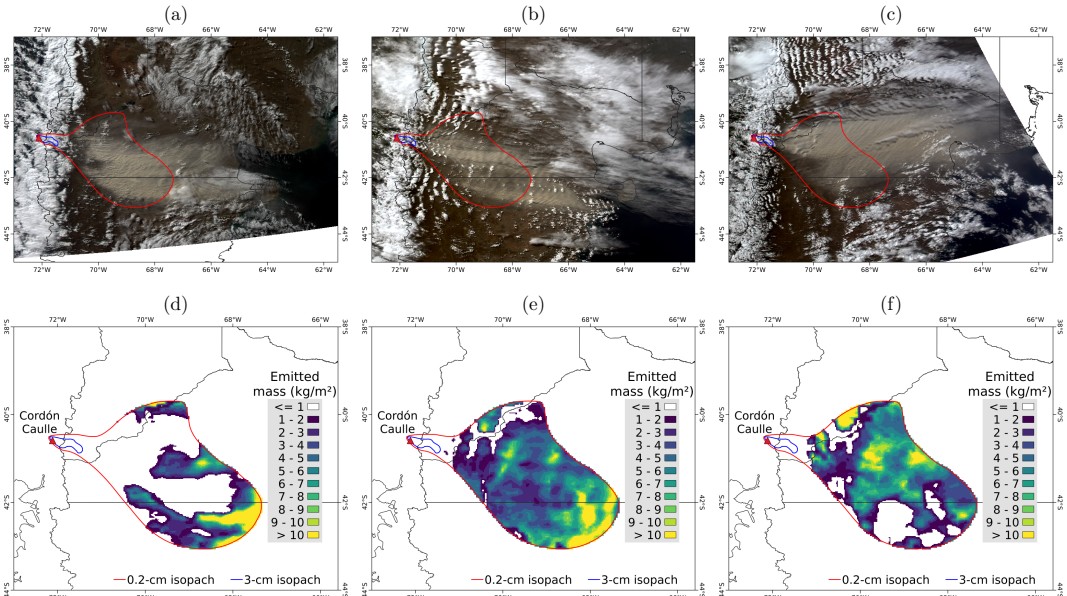

**Figure 11.** Ash resuspension over the Patagonian steppe detected by MODIS on: (**a**) 29 August 2011 at 18:15 UTC, (**b**) 24 September at 18:50 UTC and, (**c**) 15 October at 19:10 UTC. Bottom plots show the model emitted mass per unit area during (**d**) 27 August 00:00 UTC–30 August 00:00 UTC, (**e**) 24 September 00:00 UTC–26 September 00:00 UTC, and (**f**) 14 October 12:00 UTC–18 October 00:00 UTC. The external limit of the emission source area is defined by the 0.2-cm isopach of the primary tephra-fallout deposit (solid red line). This area is subdivided into two areas: a proximal area limited externally by the 3-cm isopach (solid blue line) and a distal area between the blue and red contours (see Figure 2 for more details).

The WRF-ARW fields (on a Lambert conformal projection) were bi-linearly interpolated onto a regular lon–lat grid at 0.01° of horizontal resolution in order to compute the emission flux. According to MODIS satellite imagery, emission sources were constrained in an area delimited approximately by the 0.2-cm isopach (Figure 11a–c, solid red line) and, consequently, emission was assumed to occur

only within this spatial region in our modelling strategy (Figure 11d–f). Additionally, the emission source area is subdivided into two areas: (i) a proximal area, limited by the 3-cm isopach (solid blue line); and (ii) a distal area, between the 3-cm and 0.2-cm isopachs (blue and red contours). The weighted average GSDs computed from the proximal and distal samples (see Section 2.1 and Figure S1, supplementary material) were used to characterise the proximal and distal emission areas, respectively. Both GSDs are represented in Figure S2 (supplementary material). Consequently, the distal emission area is characterised by a larger fraction of fine ash. Based on field measurements, particle densities were set to $\rho_p = 2500 \, \text{kg m}^{-3}$ (see Section 2.1), whereas a constant and uniform particle sphericity of 0.9 was assumed.

Figures 11d–f show the total mass per unit area emitted over the entire period of time simulated. Note that all emission sources are predicted over the Patagonian steppe, with no emission occurring west of the 71.4° W meridian. Surprisingly, neither the model nor the satellite images identify resuspended ash in the proximal zone around the CC. In principle, this could be explained by two main reasons: (i) the coarser granulometric characteristics of proximal deposits; and (ii) the less propitious environmental factors. In order to discriminate the effect of the proximal and distal GSD, we also simulated the three outbreaks only considering the distal GSD for the entire emission area. It was found that no proximal emission is predicted by the model regardless the granulometry. As a consequence, the inhibition of resuspension can be fully explained by the environmental conditions prevailing in proximal regions according to our modelling strategy.

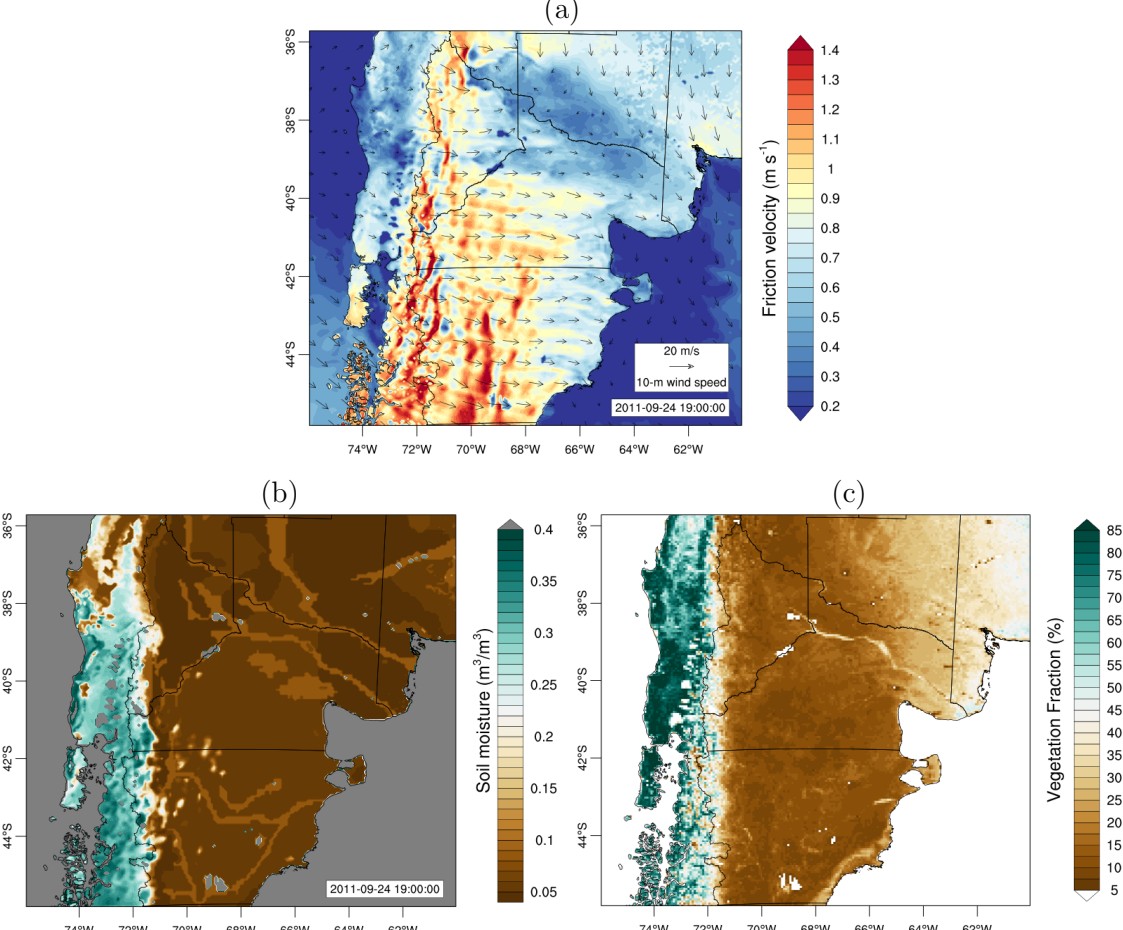

**Figure 12.** WRF fields required by the emission scheme at 6-km resolution for 24 September 2011 at 19:00 UTC: (**a**) friction velocity and 10-m wind vectors; (**b**) volumetric soil moisture; and (**c**) vegetation cover fraction. Two regions with contrasting environmental conditions for aeolian erosion can be recognised, with ideal conditions for ash resuspension across the dry regions east of the Andes.

In summary, satellite observations and numerical simulations strongly suggest that emission was significantly stronger across the drier regions of the Patagonian steppe. However, it is not possible to conclude from the results presented here that no ash resuspension episodes occurred in proximal areas at all. For example, some emission sources may be associated with scarce vegetation areas that cannot be correctly represented given the horizontal resolution used here (6 km). Higher resolution WRF-ARW simulations are expected to lead to a better description of the complex terrain associated with proximal deposits.

Temporal evolution of the total emission rate (i.e., fluxes integrated over the spatial domain) shows a strong diurnal pattern in all the studied cases (Figure 13). Ash resuspension predominates in the afternoon, with maximum emissions at around 18:00 UTC (15:00 LT). Note how the emission rate increases abruptly between 12:00 and 14:00 UTC, whereas almost no emission occurs during night-time atmospheric conditions. This emission model result is in total agreement with geostationary satellite observations and weather station reports (see Figure 8b).

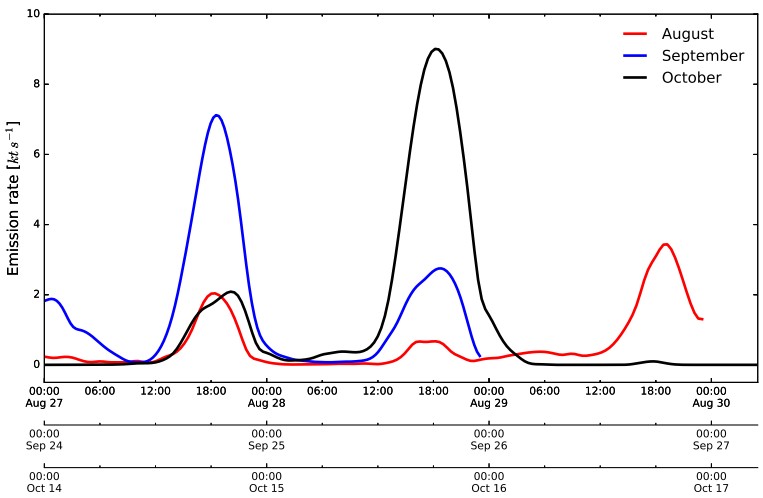

**Figure 13.** Time evolution of total emission rate in $kt\,s^{-1}$. Note the strong diurnal variability, with maxima between 18:00 and 19:00 UTC (15:00 and 16:00 LT). Emissions are strongly reduced on 28 August due to precipitation events occurring over wide areas of Patagonia during that day.

A remarkable feature of ash resuspension in Patagonia is revealed in Figure 13. In general, massive ash resuspension episodes occur as a sequence of two or three consecutive events due to travelling low-pressure systems creating windy conditions for 2 or 3 consecutive days. In order to illustrate this by way of an example, an animation of the simulated column mass for the August 2011 outbreak (see Section 4.3.2) along with sea-level pressure (contours in hPa) from WRF-ARW is provided in the supplementary material (Video S1). In particular, each peak in Figure 13 corresponds to a resuspension event, which can be recognised from satellite images (see Figure 11 and Section 4.3.2), being the small peak on 28 August (solid red line) the only exception. Cloudy conditions prevented resuspended ash from being distinguished in satellite imagery during this day.

The August outbreak represents an interesting case, with massive resuspension and precipitation occurring simultaneously. The effect of precipitation on the emission rates is illustrated in the time series of Figure 14. Clearly, ash resuspension is strongly inhibited on 28 August due to precipitation. However, the resulting increase in soil moisture was insufficient to fully inhibit ash emission during the 27 and 29 August, which is in complete agreement with satellite images. The situation was different on 28 August, when high levels of moisture remained during the afternoon, thereby preventing ash from being remobilised. According to the results for this specific case, more than 1 mm of daily accumulated precipitation was required to inhibit resuspension on 28 August. Note that windy conditions prevailed throughout the August outbreak and, therefore, the suppression of emission should be attributed to the precipitation.

4.3.2. FALL3D Results And Validation

We determined the erodibility parameter $\sigma$, defined in Equation (8), to calibrate the source strength in the emission scheme by matching modelled surface concentrations with observations. To this purpose, we estimated concentration from visibility reported at 43 weather stations using an empirical relationship between Total Suspended Particle (TSP) concentration and visibility [72]. The correlation between simulated and observed data for daily-averaged surface concentration is shown in Figure 15. The time evolution of near surface TSP concentration is also well captured by the model in most stations, as shown in the time series of Figure S3 (supplementary material). The criterion for obtaining the calibration factor was to impose the same quantity (50%) of over- and under-estimations, meaning that the ideal trend in Figure 15 (solid black line) divides the plane into two halves with equal number of data points. This resulted on $\sigma = 0.21$.

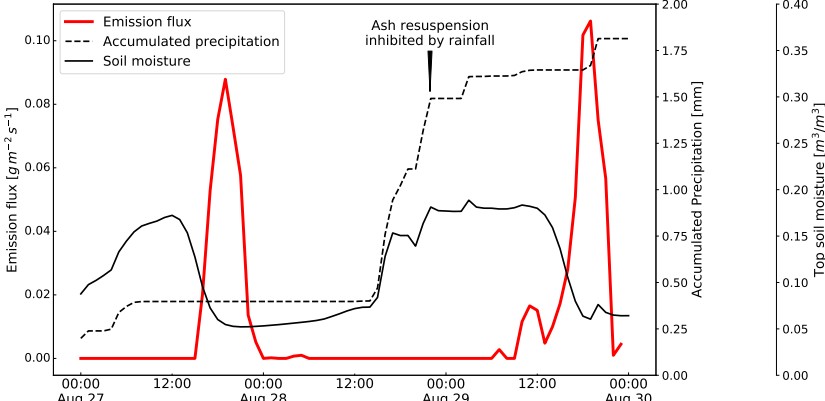

**Figure 14.** Time series of emission flux at 41.31° S, 69.72° W during the August outbreak (solid red line). The typical diurnal variability associated with resuspension events can be recognised except for 28 August, when an sharp increase in soil moisture (solid black line) due to rainfall (dashed line) prevented ash emission during the afternoon.

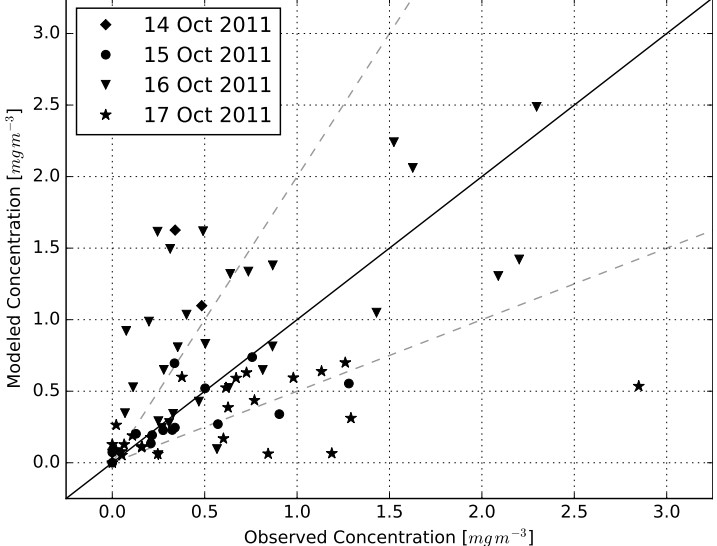

**Figure 15.** Daily-averaged Total Suspended Particle (TSP) concentration for the October outbreak. Observed concentration was derived from visibility reports at 43 weather stations. The emission scheme was calibrated to simulate concentration considering an erodibility parameter of 0.21, representing a calibration factor for the theoretical emission flux. Dashed lines indicate a 1:2 ratio between model and observations, the perfect agreement is represented by the solid black line.

Figures 16d–f show a snapshot of the simulated ash column mass for each outbreak—28 August at 06:00 UTC, 25 September at 18:00 UTC, and 15 October at 15:00 UTC, respectively. More examples are provided in Figure S4 (supplementary material). An animation of the simulated column mass for each outbreak can be found in Videos S2–S4 (supplementary material).

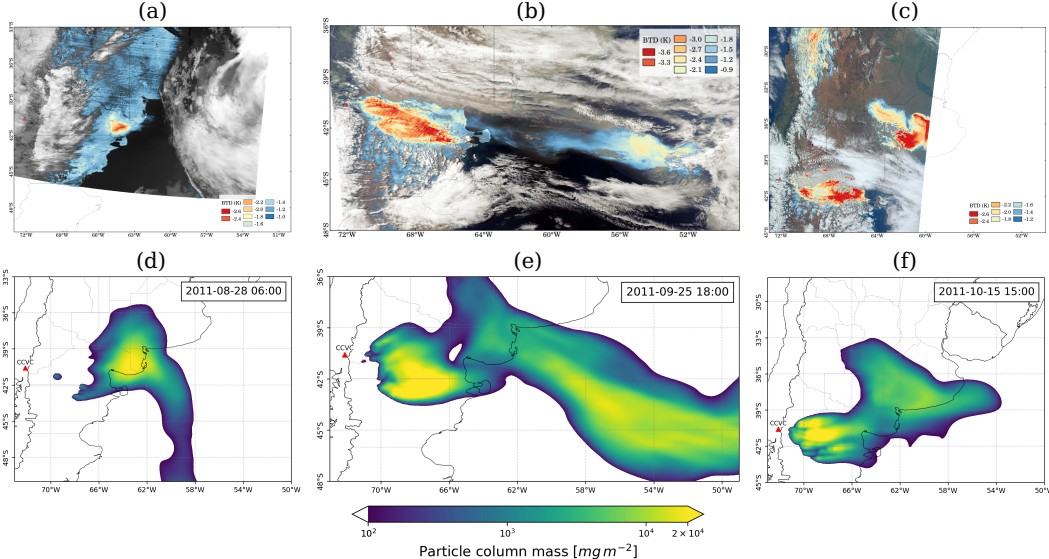

**Figure 16.** MODIS satellite imagery (**a–c**) and FALL3D column mass (**d–f**). The BTD (brightness temperature difference) method is used to detect airborne volcanic ash for: (**a**) 28 August 2011 at 05:30 UTC; (**b**) 25 September 2011 at 17:55 UTC and; (**c**) 15 October 2011 at 14:55 UTC.

MODIS satellite imagery (Figure 16a–c) can be used to further validate model results in a semi-quantitative way. To this purpose, we processed calibrated radiance from MODIS Level-1B data. Airborne ash was detected using the BTD (brightness temperature difference) method, based on the difference of brightness temperatures between the infrared channels at 10.7 and 12 μm [76]. Negative values of BTD, indicating a signal compatible with the presence of ash in the atmosphere, are represented in Figures 16a–c. In the night image of 28 August at 05:30 UTC (Figure 16a) a resuspended ash cloud was detected by satellite moving northeastward towards the Atlantic Ocean. This ash cloud corresponds to residual material resuspended during the previous day. The location of the detected ash is in agreement with the FALL3D maximum column mass, at around 40–41° S (Figure 16d). Satellite images on 25 September at 17:55 UTC and 15 October at 14:55 UTC (Figure 16b,c) also show ongoing ash resuspension taking place over semi-arid regions of northern Patagonia. Besides the ash cloud related to active sources over Patagonia, it is possible to identify an additional ash cloud which is detached from the first one and represents residual airborne ash resuspended during the previous day. In consequence, the gap between both clouds is a clear manifestation of the diurnal variability associated with the ash resuspension phenomena in Patagonia. Similarly, the WRF-ARW/FALL3D modelling system is capable of capturing two distinct events in consecutive days. Note that the secondary maximum in the simulated fields of column mass (Figure 16e–f) corresponds actually to the first event. To illustrate this point, Figure 17 shows the near surface TSP concentration at Santa Rosa weather station (36°36′ S, 64°16′ W), where ash in suspension was reported during the October outbreak. Concentration derived from visibility reports (solid red line) is compared to numerical simulation (solid black line). Star symbols represent hourly reports of suspended ash issued by the weather station.

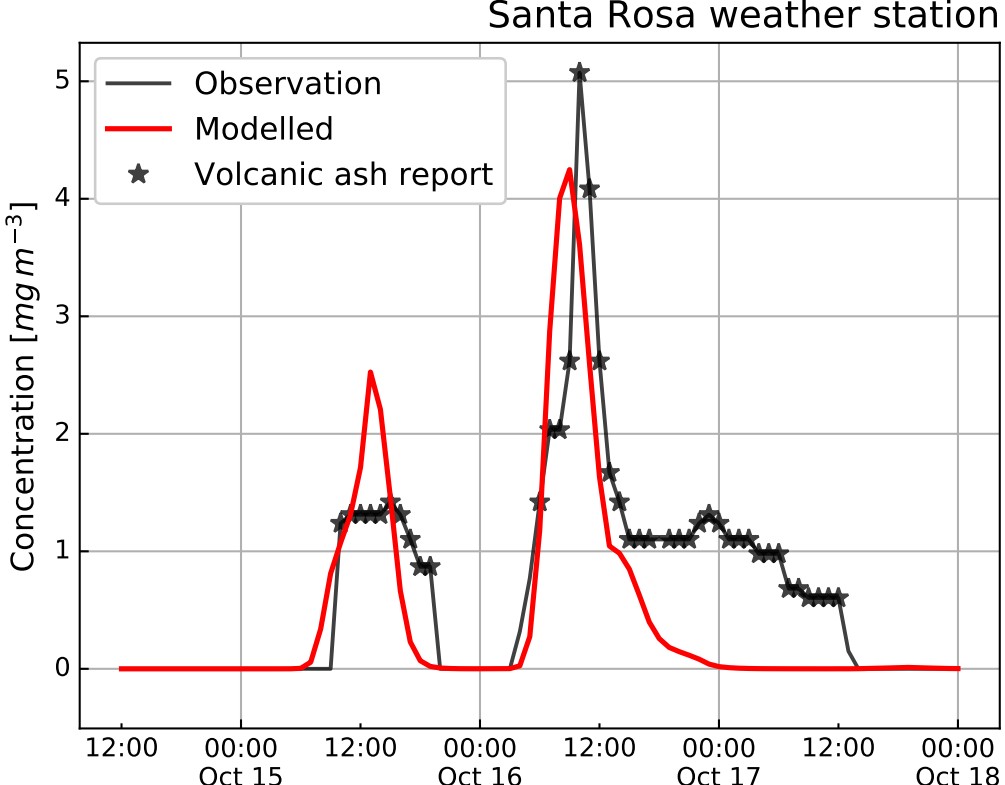

**Figure 17.** Two consecutive resuspension events detected at Santa Rosa weather station (36°36′ S, 64°16′ W) during the October 2011 outbreak. Surface concentration according to model (solid red line) and observational data (solid black line) derived from hourly reports of visibility are in good agreement. Ash-in-suspension reports are indicated by stars.

In contrast to volcanic plumes from explosive eruptions, ash clouds from resuspension occur predominantly within the lower troposphere. Layers of volcanic aerosols were detected by the space-based lidar on board the CALIPSO satellite. Figure 18a,b show CALIOP measurements of 532-nm total attenuated backscatter obtained during CALIPSO overpasses for different outbreaks. The lidar profiles on 28 August show a strong signal between the 38°S and 42°S latitudes in agreement with the MODIS images (Figure 16a). A similar vertical structure is reproduced by the model (Figure 18c), with most of the mass concentrated below of 3–3.5 km a.s.l. and a maximum top height of the ash layer at ∼41° S.

The presence of water/ice clouds on the backscatter signal acquired by the lidar on 15 October 2011 (Figure 18b) makes ash detection slightly more difficult. The region enclosed by the solid red line in Figure 18b shows the lidar signal associated with the layer of volcanic aerosols, as discussed by Ulke et al. [8], while lidar signal outside this region corresponds to water/ice clouds. The vertical cross section of modelled concentration is consistent with the spatial distribution observed except north of 41° S, possibly because the lidar signal was highly attenuated by upper meteorological clouds (Figure 18d).

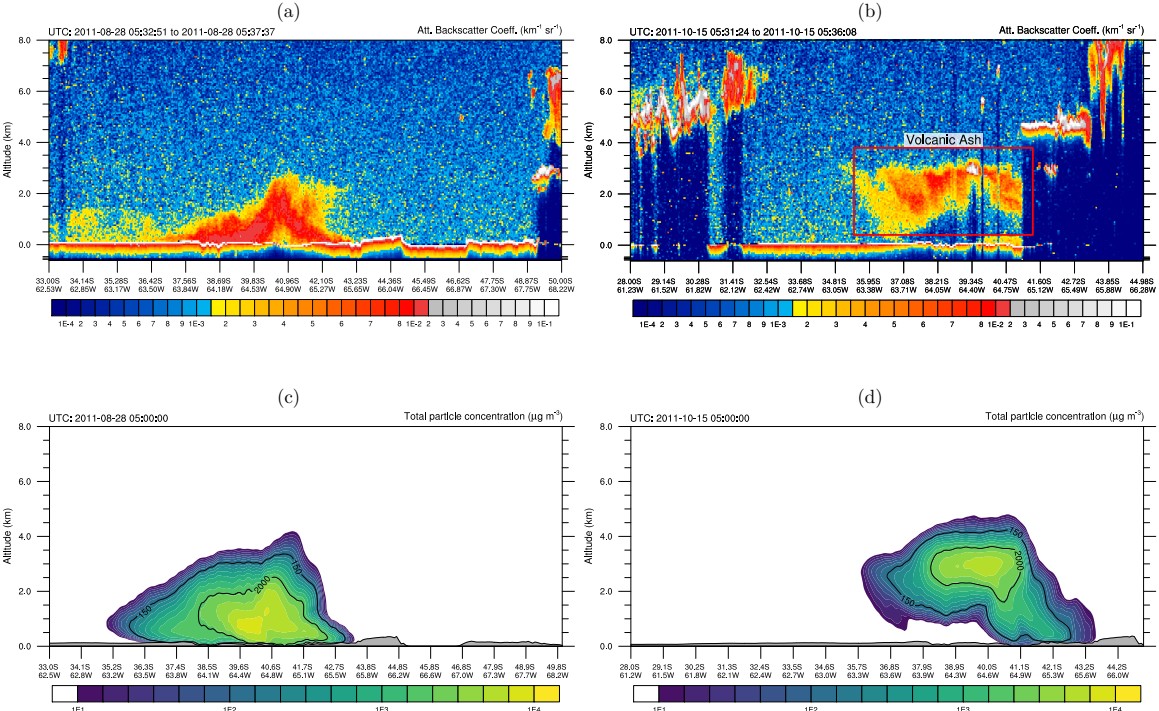

**Figure 18.** Vertical distribution of resuspended ash along CALIPSO transects. Attenuated backscatter coefficient at 532 nm on (**a**) 28 August 2011 between 05:33 and 05:37 UTC and (**b**) 15 October 2011 between 05:31 and 05:33 UTC. Model vertical cross sections of concentration are also shown for (**c**) 28 August 2011 at 05:00 UTC and (**d**) 15 October at 05:00 UTC.

## 5. Discussion

We presented numerical simulations and observation data to characterise the ash resuspension activity in northern Patagonia following the 2011 Cordón Caulle volcanic eruption. Recurrent aeolian remobilisation of the tephra-fallout deposits prolonged and exacerbated the initial impact associated with the primary volcanic activity and affected rural communities in the Patagonian steppe for years.

### 5.1. Emission Scheme

The applicability of traditional dust emission schemes to ash resuspension should be further investigated. In particular, future studies should aim at developing and validating specific emission schemes for volcanic ash based on recent experimental studies for ash resuspension [24,25]. In this work, emission flux was computed using a semi-empirical dust emission scheme based on the parameterisation proposed by Shao et al. [51,52]. According to this modelling strategy, resuspension of fine ash is triggered by the bombardment of saltating grains, that is, the emission scheme is sensitive to both fine and coarse size fractions of volcanic particles, as indicated by Equation (8). Consequently, it is important to realistically characterise the primary tephra-fallout deposit based on field data. In this work, average GSDs for proximal and distal areas were computed from proximal and distal samples.

### 5.2. Operational Implementation

The implementation of an operational system based on this modelling strategy is challenging due to the unavailability of tephra-fallout deposit data and the lack of a reliable description of soil moisture for the most superficial layer. First, the characterisation of tephra-fallout deposits requires field data which is not available in real time. In this case, deposit information could be derived from probabilistic modelling of tephra sedimentation based on historical eruptive scenarios. Second, our research has underlined the importance of incorporating realistic soil moisture information

into the model. However, soil moisture provided by most of the regional NWP models and global datasets (e.g., ERA-Interim/ERA5 reanalysis or GFS) is not representative of soil surface conditions involved in the ash emission process. In this case, a specific configuration of the land-surface model is required to obtain relevant information for the ash emission scheme.

### 5.3. Spatial Distribution of Sources

Satellite observations and numerical simulations strongly suggest that major emission sources were distributed across the Patagonian steppe. According to numerical simulations, ash resuspension was inhibited in proximal areas due to the environmental conditions prevailing in this region, even though the GSD associated with Unit III of the CC deposit may be potentially remobilised. However, it is not possible to conclude from the results presented in this study that no ash resuspension episodes occurred in proximal areas at all. Since a complex terrain is associated with proximal deposits, simulations at higher resolution are required to capture local emission sources over this region. In addition, one might wonder if dust emission schemes, typically applied to arid regions, are suitable for wetter regions such as the temperate, highland climatic area of the Nahuel Huapi National Park, characterised by an average annual precipitation of more than 800 mm [37].

### 5.4. Diurnal & Seasonal Variability

We provided further evidence of a strong diurnal and seasonal variability of the ash resuspension phenomena in Patagonia. Ash resuspension activity was found to be more intense during daytime hours in austral spring and summer according to observations at Bariloche. Modelled emission fluxes correctly reproduces this diurnal variation of emission fluxes. This regular pattern can facilitate the forecasting of resuspension events and might also be useful to distinguish resuspended ash from volcanic eruptions emissions.

### 5.5. Spatio-Temporal Distribution of Airborne Ash

Additionally, diurnal variability can be easily recognised from satellite imagery and the spatio-temporal distribution of airborne ash concentration. A good agreement was found between the simulated column mass and airborne ash detected using BTD values from MODIS satellite imagery. A strong backscattered signal measured by the CALIPSO lidar allowed to identify a layer of resuspended ash during the August and October outbreaks. In both cases, airborne particles were found in the lower layers of the atmosphere up to 3 km above sea level. Unfortunately, a direct comparison between model and observations is not possible from the presented results, as the attenuated backscatter coefficient is not directly proportional to concentration of aerosols. However, the modelled concentration contour of $150 \, \mu\text{g}\,\text{m}^{-3}$ (Figure 18c,d) follow a similar spatial distribution than the ash layer detected by the lidar. Furthermore, the model seems to slightly overestimate the top height of the ash layer.

## 6. Conclusions

A detailed study of ash resuspension events in northern Patagonia following the 2011 Cordón Caulle eruption has been presented, including a comprehensive description of the spatial distribution of emission sources, diurnal and seasonal variability, frequency of events, transport patterns, and spatio-temporal distribution of airborne ash. Comparisons between numerical simulations and observations (satellite imagery, lidar data, weather station reports) show a good agreement for the analysed events.

A novel modelling approach has been adopted to overcome some limitations of previous studies dealing with ash resuspension modelling. The main improvements include:

- A better integration of the data on primary tephra-fallout deposit (i.e., field-based GSD data from both proximal and distal samples and particle density).

- A new strategy to obtain a more realistic description of the top soil moisture of significant relevance to the ash emission scheme.
- The inclusion of the effects of vegetation cover on wind erosion.
- A new approach to determine the grain size associated with resuspended particles. In this case, the maximum size for resuspended particles depends on atmospheric conditions (i.e., friction velocity) and particle properties (i.e., particle size and density), and no arbitrary restriction was imposed on the grain size of the emitted particles (e.g., even coarse ash could be resuspended under certain conditions).

Our study provides the basis for further progress in the improvement of forecast accuracy of ash resuspension episodes. To this purpose, the new capabilities to be incorporated in the next release of FALL3D (version v8.1) will be explored in future studies—ensemble forecast and data assimilation.

**Supplementary Materials:** The following figures and videos are available online at http://www.mdpi.com/2073-4433/11/9/977/s1: Figure S1: Location map of the study area in Patagonia with sampling sites, Figure S2: Weighted average Grain Size Distribution (GSD), Figure S3: TSP concentration at multiple weather stations during the October 2011 outbreak, Figure S4: MODIS imagery and simulated column mass for the October 2011 outbreak, Video S1: Sea-level pressure and resuspended ash for the August 2011 outbreak, Video S2: FALL3D simulation for the August 2011 outbreak, Video S3: FALL3D simulation for the September 2011 outbreak, Video S4: FALL3D simulation for the October 2011 outbreak.

**Author Contributions:** Conceptualization, L.M.; Methodology, L.M., A.F., L.D. and C.B.; Software, A.F. and L.M.; Resources, L.D. and C.B.; Writing—original draft, L.M.; Writing—review and editing, L.M., A.F., L.D. and C.B.; Visualization, L.M.; Supervision, A.F.; Funding Acquisition, L.M., A.F., and C.B. All authors have read and approved the final version of the manuscript.

**Funding:** Leonardo Mingari thanks CONICET for their PhD fellowship. Lucia Dominguez was supported by the Swiss National Science Foundation (project number 200021-63152). The WRF-ARW/FALL3D modelling system has been run on the Marenostrum Supercomputer located in the Barcelona Supercomputer Center (BSC) and an HPC system installed at the National Weather Service (Argentina) with funds from the Argentinian project PIDDEF 41/10. This work has been partially funded by the H2020 Center of Excellence for Exascale in Solid Earth (ChEESE) under the Grant Agreement No. 823844.

**Acknowledgments:** We thank the Servicio Meteorológico Nacional (Argentina) for providing the meteorological data from the weather stations.

**Conflicts of Interest:** The authors declare that they have no conflict of interest.

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
