# Peer review of "Volcanic Ash Resuspension in Patagonia: Numerical Simulations and Observations"

_atmosphere, doi:10.3390/atmos11090977_

Round 1

Reviewer 1 Report

Manuscript ID: atmosphere-913780

Volcanic ash resuspension in Patagonia: numerical simulations and observations

Leonardo Mingari, Arnau Folch, Lucia Dominguez, and Costanza Bonadonna

Reviewers report          Aug 28th 21020.

General comments

The article uses the WRF model coupled with the FALL3D model to simulate Volcanic ash resuspension in Patagonia  from the 2011 Cordón 10 Caulle eruption in Chile. The background material is sufficient and includes references as needed for the two models. There are minor flaws in the research design, there are missing definitions and incomplete calibration. The methods are well described and simulation results are adequately presented. The conclusions presented are mainly the innovations in the numerical model construction, the numerical results themselves are hardly discussed in the conclusion.

There are several things that could be duscussed but are not, WRF is a well known model and the article is rather long, chapter 4 alone is 11 pages, the authors should try to cut it down.

Specific comments

  1. 6. Eq. 2. Exponent ¼ should be ½.  This eq. Is unnecessary as the velocity fluctuations cannot be defined except from the u*, that again should be defined from the velocity profile, e. g. U2 and U10 on fig. 12a or the roughness.

  1. 6. L. 201 The correction factors fw and fλ....  Undefined parameter.  Either define or refer to fig. 3.

  1. 6. L. 204 maximum amount of adsorbed water, w′....  Velocity fluctuation in Eq. 2.

  1. 7. Fig. 3 Threshold Friction Velocity - Ideal case... Comparison to Shields parameter could be interesting.

  1. 7. Fig. 3 Saltation Flux.... In this formula u* must be derived from the actual wind speed.  How ??

  1. 7. L. 224 σ represents a calibration factor...  There are many more opportunities to use the available data for calibration purposes, as to find the best values of all the factors used in the model.

Reviewer 2 Report

The paper is very interesting, thoughtful, and certainly contains valuable ideas worthy of publication. Noting grate sensitivity of the emission rates to some key parameters of the resuspension model used by the authors, a comparison of the present model results with those obtained from a simpler parameterization for ash emissions due to resuspension would be important part of the investigation. This would allow for seeing actual benefits of using the highly sophisticated emission scheme of Shao. Yet, the study is very comprehensive and informative by itself, so that it can be published without any addition improvements.

A minor bug:

Line 19 “… major emission sources were distributed …”

---- Should be changed to: “… major emission sources of resuspended ash were distributed …”, otherwise it is not clear what do the authors mean by “emission sources”. 

Also, I would recommend add into Conclusions some quantitative estimates which would show an extent to which an original TGSD of the fallout is changed by resuspension process on a time scale of a year or more. Particularly, what proportions of coarse (>63 um) and fine ash particles from the whole fallout deposit have been remobilized in the eruptions considered?

Kind regards, Reviewer

Author Response

Response to Reviewer 2 Comments

We appreciate the time and the effort that the reviewer dedicated to providing feedback on our manuscript and are grateful for the comments. We have incorporated some of the suggestions made by the reviewer. Red text below shows our point-by-point responses to the reviewer's comments and concerns.

Comments and Suggestions for Authors

The paper is very interesting, thoughtful, and certainly contains valuable ideas worthy of publication. Noting grate sensitivity of the emission rates to some key parameters of the resuspension model used by the authors, a comparison of the present model results with those obtained from a simpler parameterization for ash emissions due to resuspension would be important part of the investigation. This would allow for seeing actual benefits of using the highly sophisticated emission scheme of Shao. Yet, the study is very comprehensive and informative by itself, so that it can be published without any addition improvements.

Author response: We thank the reviewer for their positive comment. In a previous work, we showed that simpler emission schemes can be used to realistically reproduce ash resuspension events (Folch et al., 2014). However, the actual benefits of using a sophisticated emission scheme, such as the Shao scheme, would probably be a better characterization of the grain size distribution of the resuspended material. Unfortunately, this cannot be verified in the present work as we do not have GSD measurements of airborne ash. 

References:

Folch, A., Mingari, L., Osores, M. S., & Collini, E. (2014). Modeling volcanic ash resuspension--application to the 14-18 October 2011 outbreak episode in central Patagonia, Argentina. Natural Hazards & Earth System Sciences, 14(1).

 A minor bug:

Line 19 “… major emission sources were distributed …”

---- Should be changed to: “… major emission sources of resuspended ash were distributed …”, otherwise it is not clear what do the authors mean by “emission sources”. 

Author response: The suggestion given by the reviewer was incorporated in the revised manuscript.

Also, I would recommend add into Conclusions some quantitative estimates which would show an extent to which an original TGSD of the fallout is changed by resuspension process on a time scale of a year or more. Particularly, what proportions of coarse (>63 um) and fine ash particles from the whole fallout deposit have been remobilized in the eruptions considered?

Author response: Unfortunately, the timescale involved in our simulations ranges between 48 and 72 hours, approximately. Therefore, a quantitative estimate of the remobilised material on a time scale of a year or more from our numerical simulation would be extremely inaccurate, especially taking into account the strong seasonal variability of the resuspension activity in Patagonia. This topic is best addressed in two recently published papers:

Dominguez, L., Rossi, E., Mingari, L. et al. Mass flux decay timescales of volcanic particles due to aeolian processes in the Argentinian Patagonia steppe. Sci Rep 10, 14456 (2020). https://doi.org/10.1038/s41598-020-71022-w

Dominguez L, Bonadonna C, Forte P, Jarvis PA, Cioni R, Mingari L, Bran D and Panebianco JE (2020) Aeolian Remobilisation of the 2011-Cordón Caulle Tephra-Fallout Deposit: Example of an Important Process in the Life Cycle of Volcanic Ash. Front. Earth Sci. 7:343. doi: 10.3389/feart.2019.00343

Reviewer 3 Report

This is a very well written and well presented paper on ash resuspension events in northern Patagonia following the 2011 Cordón Caulle eruption that includes a comprehensive description of the spatial distribution of emission sources, diurnal and seasonal variability, frequency of events, transport patterns, and spatio-temporal distribution of airborne ash.  It offers several novel modeling approaches to overcome limitations of previous simulations of resuspension events.  It should be published in Atmosphere.

I have only two very minor suggestions.

line 42:  It might be worthwhile to spell out BSNE (Big Spring Number Eight) in "triple BSNE samplers."

line 120:  "the last decades" might read better if it were "the last three decades."

Author Response

Response to Reviewer 3 Comments

We appreciate the time and the effort that the reviewer dedicated to providing feedback on our manuscript and are grateful for the comments. We have incorporated the suggestions made by the reviewer. Red text below shows our point-by-point responses to the reviewer's comments and concerns.

Comments and Suggestions for Authors

This is a very well written and well presented paper on ash resuspension events in northern Patagonia following the 2011 Cordón Caulle eruption that includes a comprehensive description of the spatial distribution of emission sources, diurnal and seasonal variability, frequency of events, transport patterns, and spatio-temporal distribution of airborne ash.  It offers several novel modeling approaches to overcome limitations of previous simulations of resuspension events.  It should be published in Atmosphere.

I have only two very minor suggestions.

line 42:  It might be worthwhile to spell out BSNE (Big Spring Number Eight) in "triple BSNE samplers."

Author response: The suggestion given by the reviewer was incorporated in the revised manuscript.

line 120:  "the last decades" might read better if it were "the last three decades."

Author response: The suggestion given by the reviewer was incorporated in the revised manuscript.